# Externally induced frontoparietal synchronization modulates network dynamics and enhances working memory performance

Ines R Violante[1,2]*, Lucia M Li[1], David W Carmichael[3], Romy Lorenz[1], Robert Leech[1], Adam Hampshire[1], John C Rothwell[2], David J Sharp[1]

[1]The Computational, Cognitive and Clinical Neuroimaging Laboratory, Department of Medicine, Imperial College London, London, United Kingdom; [2]Sobell Department of Motor Neuroscience and Movement Disorders, UCL Institute of Neurology, University College London, London, United Kingdom; [3]Developmental Imaging and Biophysics Section, UCL Institute of Child Health, University College London, London, United Kingdom

**Abstract** Cognitive functions such as working memory (WM) are emergent properties of large-scale network interactions. Synchronisation of oscillatory activity might contribute to WM by enabling the coordination of long-range processes. However, causal evidence for the way oscillatory activity shapes network dynamics and behavior in humans is limited. Here we applied transcranial alternating current stimulation (tACS) to exogenously modulate oscillatory activity in a right frontoparietal network that supports WM. Externally induced synchronization improved performance when cognitive demands were high. Simultaneously collected fMRI data reveals tACS effects dependent on the relative phase of the stimulation and the internal cognitive processing state. Specifically, synchronous tACS during the verbal WM task increased parietal activity, which correlated with behavioral performance. Furthermore, functional connectivity results indicate that the relative phase of frontoparietal stimulation influences information flow within the WM network. Overall, our findings demonstrate a link between behavioral performance in a demanding WM task and large-scale brain synchronization.

*For correspondence: i.violante@ imperial.ac.uk

## Introduction

Cognitive processes depend on coordinated interactions among large-scale distributed brain networks. Flexible and rapid propagation of information across distant brain regions is necessary to support these functions. A prominent hypothesis is that information exchange within and between networks occurs through the oscillatory synchronization of neuronal activity, such that synchronous neuronal firing binds neurons into ensembles engaged in specific computational functions (*Varela et al., 2001*; *Fries, 2005*; *Womelsdorf et al., 2007*; *Fries, 2015*; *Parkin et al., 2015*). Rhythmic synchrony increases network efficiency, a process thought to be particularly relevant for demanding cognitive processes such as working memory (WM) (*Fries, 2005*; *Deco et al., 2011*; *Fell and Axmacher, 2011*; *Fries, 2015*; *Constantinidis and Klingberg, 2016*).

Frontal and parietal brain regions support WM (*Cohen et al., 1997*; *Prabhakaran et al., 2000*; *Pessoa et al., 2002*; *Todd and Marois, 2004*). Oscillatory activity in the theta range (4–8 Hz) appears to organize local neuronal ensembles across distant regions during WM processes (*Buzsáki, 1996*; *Sarnthein et al., 1998*; *Rutishauser et al., 2010*). Theta power increases during complex

**eLife digest** Like an orchestra that relies on the coordinated efforts of its members, the brain depends on its many regions working together to perform tasks such as memorizing a phone number or the name of someone we recently met. Many areas in the brain that are involved in these processes are located far apart, and so performing these tasks efficiently depends on the regions being able to communicate and coordinate information. Rhythmic waves of electrical activity in the brain are considered to be essential to maintain the flow of information. These brain waves occur when many brain cells repeatedly send signals at the same time, and the precise 'beat' of the signals might be especially important when performing more complex tasks.

In recent years, cheap and safe electrical stimulation has shown promise in being able to influence brain waves. This technique can be used to investigate the importance of precise timing between brain waves and its impact on performing tasks, as well as changes in brain activity that occur when tasks are more complex. For example, is a person's behaviour affected if electrical stimulation is used to make their brain waves more or less synchronized?

Violante et al. stimulated distant regions of the brain of volunteers and monitored how they performed in tasks with varying difficulty. When these regions were stimulated with the same 'beat' to make the brain waves more synchronized, the person performed better in the more difficult tasks. In these tasks, participants had to monitor number sequences, and spot repeated patterns in the remembered information. To better understand this effect Violante et al. performed brain stimulation while collecting brain scans. The scans showed that stimulation at the same 'beat' increases brain activity in the regions involved in task performance and changes the pattern of how regions communicate in the brain. This finding suggests that the 'beat' of brain waves is important for task performance, and that stimulating the brain externally can alter how regions in the brain communicate.

Future studies could extend these findings to patients, particularly those with the kind of damage to their brain that slows the communication between its distant regions. Stimulating these patients' brains could help bypass the internal delays, and help them to complete everyday tasks more efficiently.

manipulations of items in WM (*Sauseng et al., 2005*), with increasing memory load (*Jensen and Tesche, 2002*; *Payne and Kounios, 2009*), and is correlated with WM performance (*Jacobs et al., 2006*; *Fuentemilla et al., 2010*). Moreover, the relative phases of these oscillations influence encoding and retrieval success (*Rizzuto et al., 2006*). For example, frontoparietal theta synchrony with ~0° phase lag is associated with the maintenance and manipulation of information in WM (*Sauseng et al., 2005*; *Polanía et al., 2012*).

With the aim of providing causal evidence for the role of oscillatory synchronization during demanding WM performance, as well as to investigate the underlying neural mechanisms by which frontoparietal phase synchronization influences verbal WM, we conducted two experiments using transcranial alternating current stimulation (tACS) to selectively entrain endogenous brain rhythms (*Antal and Paulus, 2013*). In- and anti-phase theta frequency (6 Hz) stimulation was applied across two key nodes of the right frontoparietal network: the middle frontal gyrus and inferior parietal lobule. Similar tACS stimulation has previously been shown to produce phase-dependent modulations of WM performance (*Polanía et al., 2012*).

In this study, we conducted two complementary experimental approaches. In Experiment 1 we used conventional continuous theta stimulation to investigate the role of frontoparietal phase synchronization on cognitive performance. Cognitive demands were varied by using a combination of choice reaction and N-back tasks of increasing difficulty. This allowed us to test the prediction that tACS effects would only be seen when cognitive demands were high. The right hemisphere was stimulated because increasing demand in verbal N-back conditions was associated with stronger activity and effective connectivity within the right hemisphere WM network (*Fedorenko et al., 2013*; *Dima et al., 2014*), as well as a more pronounced effect of tACS on WM performance (*Jaušovec et al., 2014*). In Experiment 2 we investigated the neurobiological substrate of the phase

dependent effects of tACS. Our fMRI-tACS experimental design used short trains of tACS that avoid aftereffects of the stimulation (*Vossen et al., 2015*) whilst maximizing power to detect changes in the blood oxygen level-dependent (BOLD) signal. Functional (f) MRI is well suited to this purpose because BOLD: (1) provides a measurement of brain activity during tACS without complex artifacts that can severely contaminate traditional electroencephalography (EEG) and magnetoencephalography (MEG) recordings (*Cabral-Calderin et al., 2016*; *Vosskuhl et al., 2016*; *Noury et al., 2016*); and (2) is unaffected by the phase of the applied stimulation.

We tested the following hypotheses: (1) behavioral WM performance is enhanced by entraining synchronous activity within a right frontoparietal network activated by WM demands; (2) regional brain activity in this frontoparietal network is modulated in a phase dependent manner; (3) changes in BOLD induced by phase-dependent network stimulation correlate with behavioral performance; (4) functional connectivity across distant brain regions is modulated in a phase-dependent manner, in agreement with computational and empirical evidence that phase synchronization enables flexible reconfiguration of information flow between brain areas (*Womelsdorf et al., 2007*; *Akam and Kullmann, 2014*). We found that synchronous (in-phase) stimulation caused an improvement in performance and led to increased frontoparietal activity when cognitive demands were high. These results show that external manipulations of large-scale brain synchronization can be used to shape activity and connectivity within brain networks, and those modulations are detectable using fMRI. We additionally demonstrate the important influence of underlying cognitive demand on the neural effects of tACS.

## Results

### Synchronous frontoparietal tACS improved working memory performance

In Experiment 1, we investigated the effects of tACS on the performance of two tasks with varying cognitive loads: a simple choice reaction time task (CRT) and a verbal N-back task with 1-back and 2-back difficulty levels (*Figure 1A*). Theta (6 Hz) tACS was applied to the middle frontal gyrus and inferior parietal lobule nodes (center electrode locations at F4 and P4) of the frontoparietal network in synchronous (0° relative phase) and desynchronous (180° relative phase) conditions (*Figure 1—figure supplement 1* for electrical field distributions associated to which condition). Ten healthy volunteers performed a session of each tACS condition (synchronous, desynchronous and sham) in a pseudo-randomized single-blinded cross-over design. The effect of tACS stimulation on reaction time (RT, mean correct responses) and accuracy (percentage of correct responses) was analysed using repeated-measures ANOVAs, with task (2-back, 1-back and CRT) and tACS condition (synchronous, desynchronous and sham) as within-subjects factors.

For RTs, there was a significant main effect of task ($F_{(2,18)}$=45.47, p<0.001, $\eta_p^2$=0.84), tACS condition ($F_{(2,18)}$=8.02, p=0.003, $\eta_p^2$=0.47) and task x tACS condition interaction ($F_{(4,36)}$=3.20, p=0.024, $\eta_p^2$=0.26). This interaction was investigated in subsequent ANOVAs performed per task condition. Synchronous stimulation improved performance on the cognitively demanding 2-back task. A main effect of tACS condition on RTs was observed ($F_{(2,18)}$=7.65, p=0.004, $\eta_p^2$=0.46). This reflected faster responses in the synchronous compared to both the desynchronous ($t_{(9)} = -5.72$, p<0.001, *d* = 0.76) and sham conditions ($t_{(9)} = -2.43$, p=0.038, *d* = 0.58) (*Figure 1C*). In contrast, there was no difference between the RTs in the desynchronous and sham conditions ($t_{(9)} = 0.76$, p=0.468, *d* = 0.15). These results partly replicate recent experiments showing that performance improves with similar synchronous stimulation, but also reporting impairment in the desynchronous condition (*Polanía et al., 2012*). There was no effect of stimulation on RTs for the less demanding 1-back ($F_{(2,18)}$=1.11, p=0.351, $\eta_p^2$=0.11) and CRT ($F_{(2,18)}$=1.50, p=0.251, $\eta_p^2$=0.14) conditions, supporting a role for phase synchronization during demanding WM processes and ruling out a simple motor effect.

To investigate the extent to which synchronous stimulation improved performance in the 2-back condition, we compared RTs between the 2-back and the less demanding 1-back condition. While the expected slowing of RTs was observed for 2-back relative to 1-back performance in desynchronous and sham conditions (paired t-test between 1-back and 2-back: desynchronous tACS: $t_{(9)} =$

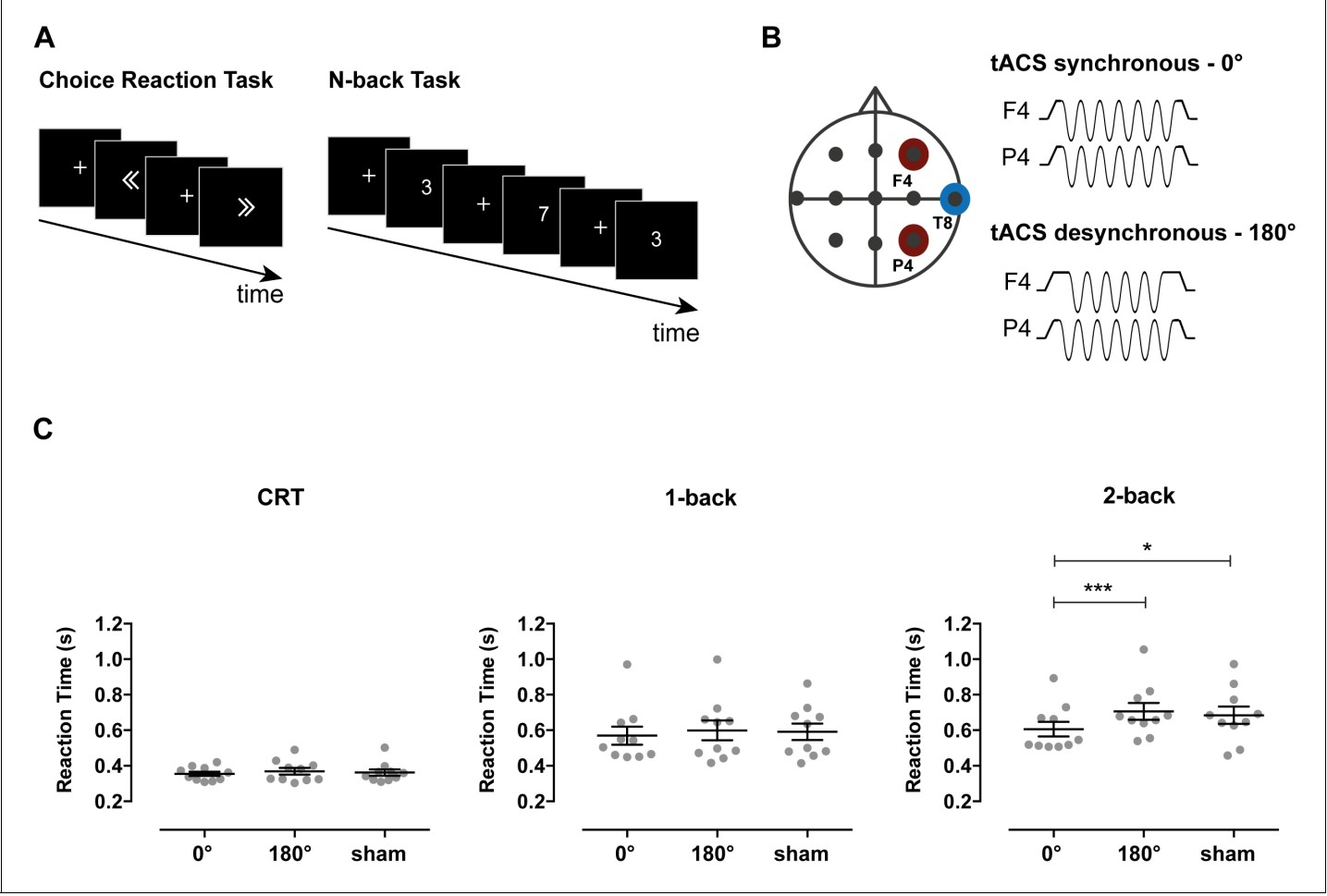

**Figure 1.** Stimulation parameters, behavioral tasks and RTs in Experiment 1. (A) Participants performed the Choice Reaction Task (CRT) and 1-back and 2-back versions of the N-back task. In the CRT task, participants were shown left or right pointing arrows and were asked to press a button as quickly and accurately as possible to indicate the direction of the arrow. In the 1-back and 2-back tasks, participants were shown a single digit number (0–9) sequentially and were required to report a repetition of the digit occurring one or two trials before, respectively. (B) tACS electrode set-up. Electrodes were positioned at frontoparietal locations F4 (middle frontal gyrus) and P4 (inferior parietal lobule) with a common return at T8 (middle temporal gyrus). TACS was applied at 6 Hz frequency with 0° relative phase between F4 and P4 in the synchronous condition and with 180° relative phase in the desynchronous condition. Stimulation was applied for the duration of the task except for the sham condition where tACS was applied for 30 s in the synchronous condition at the beginning of the task. (C) Reaction times for the CRT, 1-back and 2-back tasks for each of the stimulation conditions (n = 10). Synchronous tACS improved reaction times (RTs) for the more demanding 2-back condition compared both to desynchronous tACS and sham. Error bars represent SEM; *p<0.05, ***p<0.001.

The following source data and figure supplement are available for figure 1:

**Source data 1.** This table contains the mean reaction time of each participant for each condition in *Figure 1C*.

**Figure supplement 1.** Simulation of the electric field distributions for each tACS condition.

−4.19, p=0.002; sham: $t_{(9)}$ = −3.41, p=0.008), there was no significant difference in performance when the synchronous stimulation was applied ($t_{(9)}$ = −1.88, p=0.093).

Accuracy was unaffected by stimulation, as no main effect of tACS condition ($F_{(2,18)}$=1.01, p=0.383, $\eta_p^2$=0.10) or task x tACS condition interaction ($F_{(4,36)}$=0.51, p=0.730, $\eta_p^2$=0.05) was observed. As expected, the repeated-measures ANOVA resulted in a main effect of task condition ($F_{(2,18)}$=15.20, p<0.001, $\eta_p^2$=0.63), with the 2-back condition showing lower accuracy than the 1-back and CRT conditions (2-back: 83.9% ± 3.7%; 1-back: 95.6% ± 1.5%; CRT: 96.3% ± 0.7%).

## tACS modulated brain activity and connectivity in a tACS-phase and brain state dependent manner

In Experiment 2 we use simultaneous tACS-fMRI to investigate the neural correlates of the phase dependent effects of tACS. TACS was employed as a causal 'physiological probe' in an experimental design optimized to detect neurophysiological changes associated with the external stimulation. This approach is commonly applied in studies combining transcranial magnetic stimulation (TMS) with fMRI (*Driver et al., 2009*; *Fedorenko et al., 2013*) or EEG (*Morishima et al., 2009*), where BOLD responses are measured in a condition-dependent manner in the absence of behavioral changes.

Twenty-four participants underwent a session of simultaneous tACS-fMRI (*Figure 2A*) in which they performed separate runs of blocked 2-back and CRT tasks, for both the synchronous (0° relative phase) and desynchronous (180° relative phase) tACS conditions (N = 21 for 2-back task and N = 20 for CRT task, see Materials and methods for exclusion criteria). To measure the physiological effects of tACS, stimulation was delivered during task or rest blocks and balanced with non-tACS blocks in

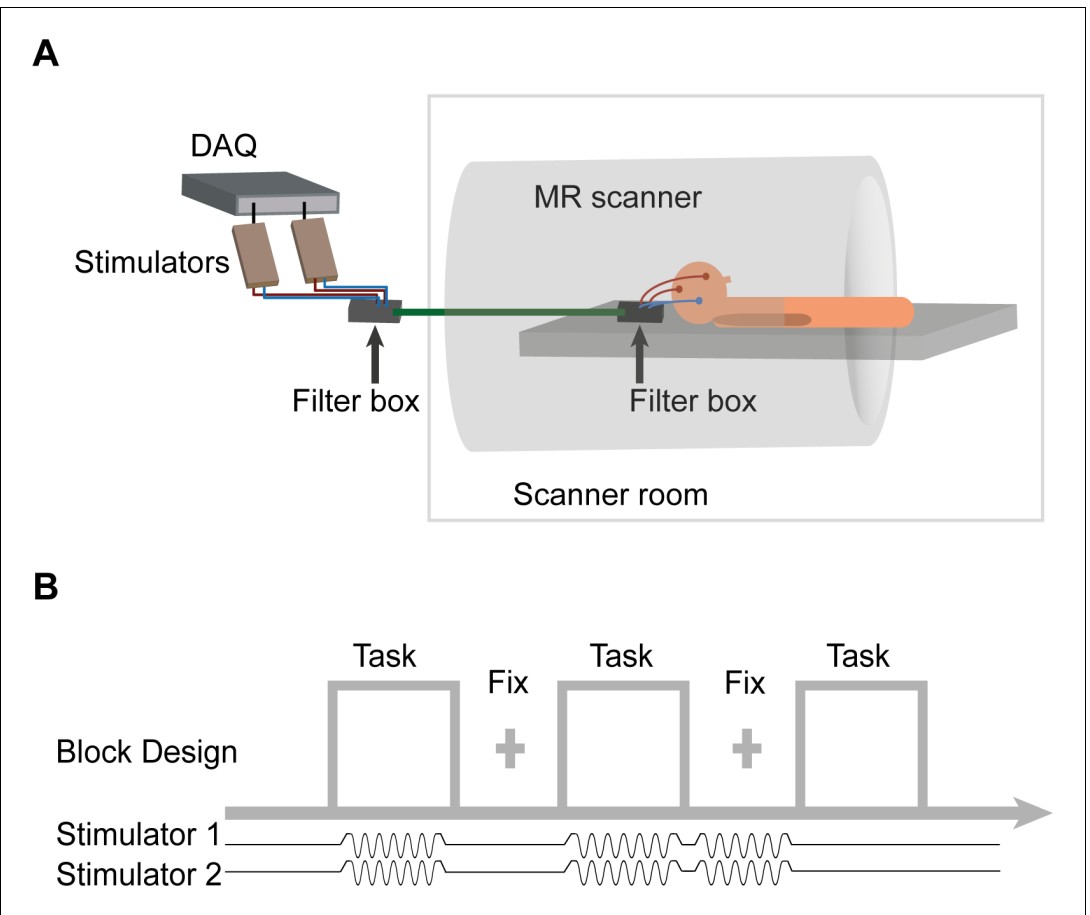

**Figure 2.** FMRI-tACS setup and stimulation protocol for Experiment 2. (**A**) The two stimulators were controlled and monitored through a digital to analog converter (DAQ). The stimulators were placed outside the MR shielded room and the current was delivered into the scanner room after being filtered from RF noise by two filter boxes. (**B**) In Experiment 2, tACS was applied in a pseudo-randomized order during Task blocks (30s) and fixation (Fix) blocks (20 s). The figure shows tACS applied at the synchronous condition (6 Hz, 0° relative phase between electrodes F4 and P4). The application of short trains of tACS was chosen to investigate the neurophysiological correlates of tACS manipulations on BOLD activity and connectivity. Each participant performed one fMRI run of each task (CRT and 2-back) and tACS condition (synchronous and desynchronous). The order of the four runs was counterbalanced across participants. In each run participants performed 24 blocks (that is,, six blocks/condition: 'task' + 'tACS ON'; 'task' + 'tACS OFF'; 'fixation' + 'tACS ON'; 'fixation' + 'tACS OFF').

a pseudo-randomized order (*Figure 2B*). The 2-back task was chosen because the performance was sensitive to tACS and the CRT as it involves low frontoparietal recruitment, allowing us to investigate how tACS interacts with cognitive state.

## Behavioral analyses

TACS stimulation did not significantly impact behavior across stimulation conditions, consistent with its use as a physiological probe (*Driver et al., 2009*; *Fedorenko et al., 2013*). As in Experiment 1, repeated-measures ANOVAs were performed on RTs and accuracy. There was no significant main effect of tACS condition on RTs or accuracy, or interaction between task (2-back and CRT) and tACS condition (all F values < 1, all p values > 0.4). As expected, there was a main effect of task on RTs ($F_{(1,18)}$=73.16, p<0.001, $\eta_p^2$=0.80) and accuracy ($F_{(1,18)}$=81.68, p<0.001, $\eta_p^2$=0.82), driven by slower responses (2-back: 0.66 ± 0.24 s; CRT: 0.44 ± 0.1 s) and reduced accuracy (2-back: 80.2% ± 1.7%; CRT: 97.1% ± 0.8%) in the 2-back than in the CRT task conditions, attributable to the expected differences in task difficulty.

## Task evoked activity

Whole-brain analysis of task evoked activity showed the expected patterns of brain activity associated with task performance, either with or without tACS stimulation (*Figure 3A*). 2-back performance was associated with increased BOLD signal in visual cortex, caudate, thalamus, cerebellum, lateral premotor cortex, dorsal cingulate and medial premotor cortex, dorsolateral and ventrolateral prefrontal cortex, frontal poles and medial and lateral posterior parietal cortex (*Figure 3A*, left). CRT performance was associated with increased activity within bilateral sensory motor regions, as well as in the supplementary motor area, visual cortex, thalamus and the putamen (*Figure 3A*, right). As expected, the activity in the frontoparietal networks was higher for the 2-back than for the CRT task. Deactivation of the default mode network (DMN) regions was observed for both the CRT and 2-back tasks compared to rest, with extensive deactivation in the ventromedial prefrontal cortex and frontal orbital cortex seen during 2-back performance (*Figure 3A*).

## Synchronous tACS locally increased task-related BOLD responses

We next assessed whether theta tACS applied synchronously or desynchronously modulates brain activity in a right frontoparietal network engaged by WM performance. A region-of-interest (ROI) analysis was performed on regions directly beneath the tACS electrodes that is, the middle frontal gyrus (MFG) and inferior parietal lobule (IPL) (see Materials and methods for details on the ROI definition).

Task-related BOLD responses were increased in the IPL and MFG regions for the synchronous tACS condition (tACS 0° > tACS OFF) selectively during performance of the 2-back task (*Figure 3B*), but not during the desynchronous tACS condition (tACS 180° > tACS OFF). As synchronous tACS improved 2-back RTs in Experiment 1, we investigated whether the modulation of activity in the IPL/MFG related to behavior. A negative correlation was observed between participants' mean BOLD signal in the IPL-electrode region and their mean RTs during synchronous tACS ($r_{(21)}$= −0.39, p=0.039, one-tailed), indicating that across participants increased brain activity related to reduced RTs (*Figure 4B*). No correlation was observed for the MFG-electrode region ($r_{(21)}$= −0.29, p=0.103, one-tailed).

During performance of the CRT task, the BOLD response was increased only for the synchronous tACS condition (tACS 0° > tACS OFF) and exclusively in the MFG-electrode region (*Figure 3C*). No correlation was observed between the increased BOLD response in the MFG and RTs during synchronous tACS.

Furthermore, to test whether the measured effects of tACS on brain activity could be explained by a simple direct effect of the electric current on the BOLD signal, we investigated activity in the brain region corresponding to the return electrode (electrode at T8, over the right middle temporal gyrus, *Figure 1A*). There was no difference in BOLD signal for this region in any of the tACS or task conditions. Similarly, during fixation periods with tACS stimulation, neither the synchronous nor the desynchronous tACS conditions elicited differences in BOLD signal compared to baseline (fixation periods without stimulation), for either the 2-back or the CRT tasks.

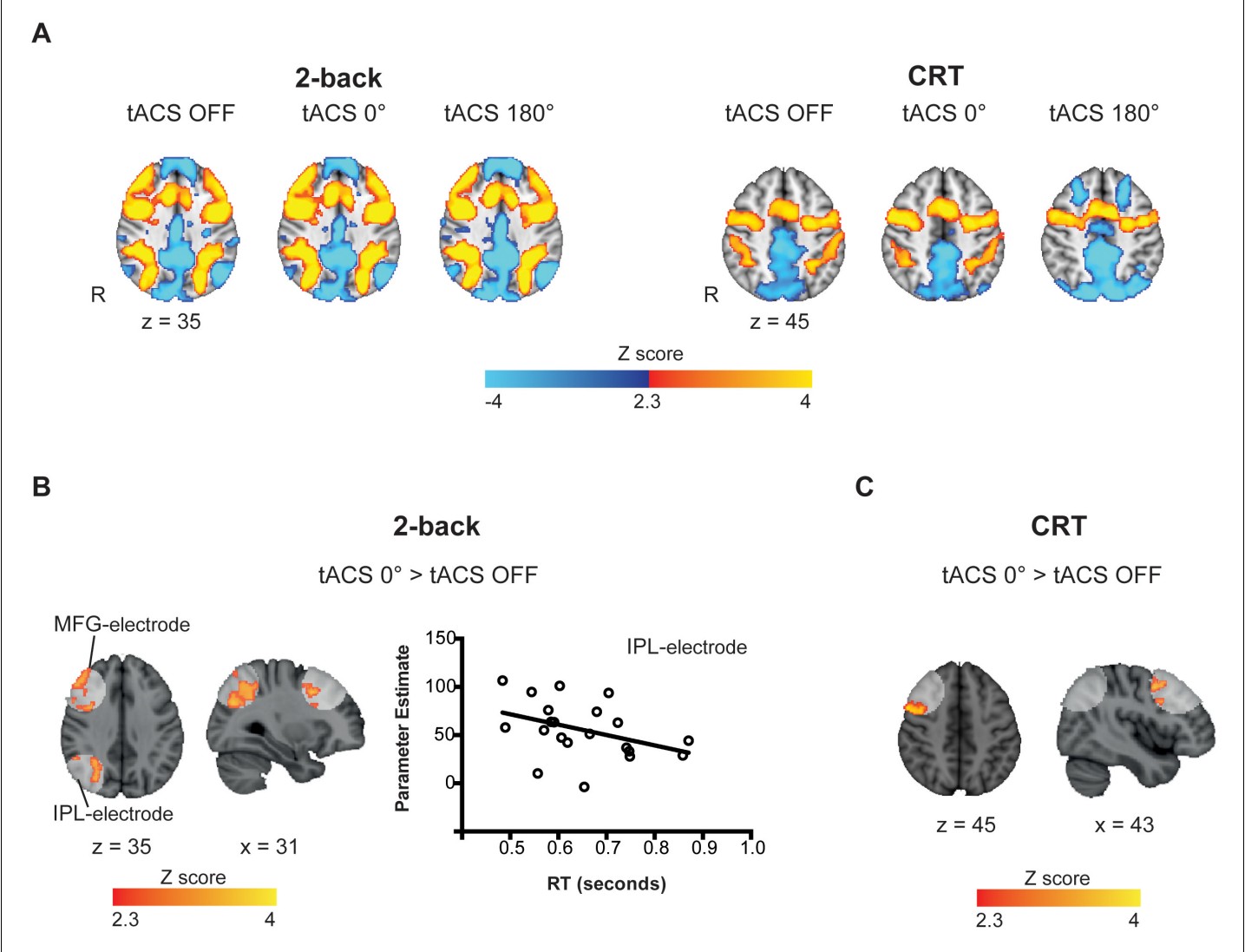

**Figure 3.** Task-related and tACS ROIs BOLD activity. (**A**) Activation patterns elicited by the 2-back (n = 21) and CRT (n = 20) tasks for tACS OFF, synchronous (tACS 0°) and desynchronous (tACS 180°) conditions. Warm colors represent increased BOLD signal relative to baseline (task > fixation tACS OFF) and cold colors a decrease in activity (fixation tACS OFF > task). (**B**) Increased activity in the IPL-electrode region is correlated with faster reaction times (RT) in the 2-back task condition (r = −0.39, p=0.039, one-tailed, n = 21). Y axis corresponds to the contrast estimates for the tACS 0° condition (extracted from the voxels showing increased brain activity for tACS 0° > tACS OFF, (**A**)) and the X axis corresponds to RTs during the synchronous tACS 0° condition. (**C**) During the CRT task, BOLD signal was increased in the MFG-electrode region for the synchronous tACS condition compared to task periods without tACS (n = 20). White regions indicate the tACS electrodes masks (tACS ROIs). Images are in the Montreal Neurological Institute (MNI) space coordinates and in radiological space. R=right hemisphere. Threshold *Z* > 2.3 with a corrected cluster significance level of p<0.05.

The following source data is available for figure 3:

**Source data 1.** This folder contains the MRI contrast maps in *Figure 3A*, both thresholded (that is, corrected for multiple comparison using cluster correction) and non-thresholded.

**Source data 2.** This folder contains the MRI contrast maps in *Figure 3B*, both thresholded (that is, corrected for multiple comparison using cluster correction) and non-thresholded and the IPL-electrode and MFG-electrode masks.

**Source data 3.** This table contains the mean reaction time and BOLD parameter estimates used for the correlation in *Figure 3B*.

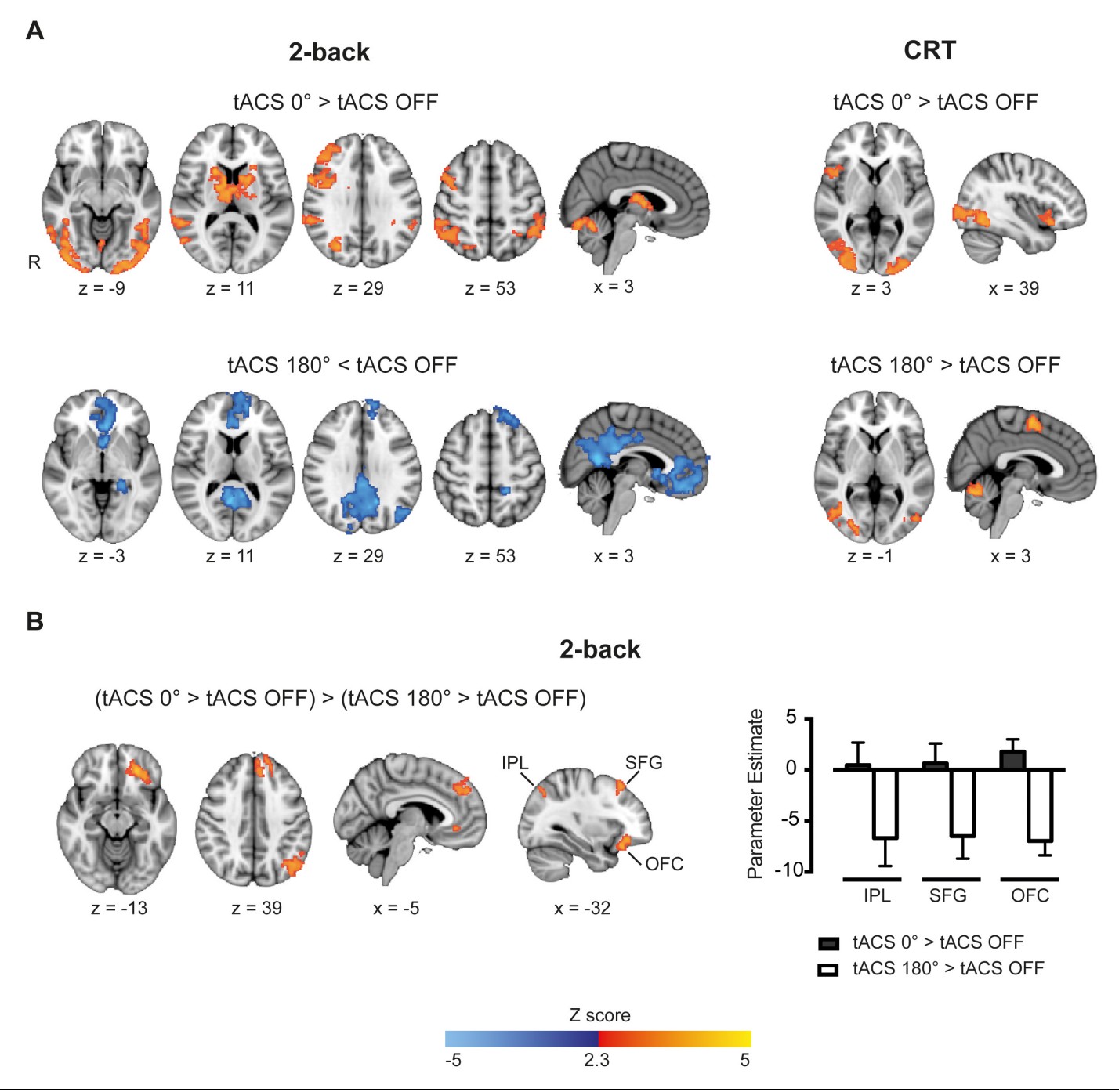

**Figure 4.** Whole-brain patterns elicited by synchronous and desynchronous tACS conditions. (**A**) Synchronous tACS during the 2-back task increased BOLD signal in brain regions underneath the parietal, IPL-electrode (inferior parietal lobule), and frontal, MFG-electrode (medial frontal gyrus), electrodes (n = 21). (**B**) During the 2-back task synchronous tACS showed significantly greater activity than tACS OFF in the right frontoparietal network and regions normally activated by task performance. Desynchronous tACS resulted in decreased activation compared to tACS OFF in default mode network regions. TACS applied during the CRT task resulted in greater activity in occipito-temporal regions for both synchronous and desynchronous tACS conditions. (**C**) Increased BOLD signal for synchronous compared to desynchronous tACS conditions, for tACS stimulation relative to no tACS stimulation, was observed in the left hemisphere in the inferior parietal lobule (IPL), superior frontal gyrus (SFG) and orbitofrontal cortex (OFC). This difference in brain activity is explained by decreased activity during desynchronous compared to synchronous tACS, right plot. Images are in the Montreal Neurological Institute (MNI) space coordinates and in radiological space. R=right hemisphere. All images have been thresholded with FSL clusterwise correction Z > 2.3, p<0.05.

*Figure 4 continued on next page*

*Figure 4 continued*

The following source data and figure supplement are available for figure 4:

**Source data 1.** This folder contains the MRI contrast maps in *Figure 4*, both thresholded (that is, corrected for multiple comparison using cluster correction) and non-thresholded.

**Source data 2.** This folder contains the MRI contrast maps in *Figure 4—figure supplement 1*, both thresholded (that is, corrected for multiple comparison using cluster correction) and non-thresholded.

**Source data 3.** This table contains the parameter estimates for each subject in the bar plot in *Figure 4B*.

**Figure supplement 1.** Whole-brain BOLD signal elicited by synchronous and desynchronous tACS conditions during Fixation.

## Synchronous tACS modulated activity across large-scale networks when cognitive demands were high

Synchronous tACS applied during performance of the 2-back task had a widespread effect on regions normally activated by task performance (tACS 0° > tACS OFF). Neural activity increased in the right frontoparietal network, with peaks of increased BOLD activity seen in the right lateral prefrontal cortex, middle frontal gyrus and bilaterally in the posterior parietal cortices Additionally, BOLD activity increases were seen bilaterally in the thalamus, basal ganglia, occipital cortex and inferior temporal gyrus (*Figure 4A* and *Supplementary file 1*, table supplement 1). A distinct pattern was observed for the desynchronous tACS condition. Increased deactivation (tACS 180° < tACS OFF) was observed in parts of the default mode network (DMN), namely the posterior cingulate cortex, precuneus, ventromedial PFC, left orbitofrontal cortex and left parahippocampal gyrus (*Figure 4B* and *Supplementary file 1*, table supplement 2). We directly contrasted synchronous and desynchronous stimulation using the contrast (tACS 0° > tACS OFF) > (tACS 180° > tACS OFF). This revealed differences induced by the two tACS phase conditions in the left hemisphere inferior parietal lobule (IPL), superior frontal gyrus (SFG) and orbitofrontal cortex (OFC), *Figure 4C*, explained by increased deactivation for the desynchronous compared to the synchronous tACS condition (*Figure 4C* and *Supplementary file 1*, table supplement 3).

The less demanding CRT task resulted in similar increases in BOLD activity in the occipital cortex for both synchronous and desynchronous tACS. Additionally, increased BOLD signal was observed for the supplementary motor cortex during the desynchronous tACS condition and right frontal orbital/insular cortex for the synchronous condition (*Figure 4B* and *Supplementary file 1*, tables supplement 4 and 5). No differences in the contrasts for synchronous vs desynchronous stimulation were observed for the CRT task.

These results show that externally inducing synchronization or desynchronization of the right frontoparietal network modulates activity across large-scale networks in a cognitive state dependent manner, such that phase dependent effects were observed clearly for the 2-back task, when cognitive demands were high. This was further confirmed during fixation periods with simultaneous tACS stimulation. For this condition, we observed no increases in the BOLD signal compared to baseline (fixation periods without tACS). Increased deactivation was observed in a number of brain regions, but there were no phase dependent differences during the CRT or 2-back runs (*Figure 4—figure supplement 1*).

## TACS modulated functional connectivity across large-scale networks

We next investigated whether tACS modulates functional connectivity (FC) within brain networks using psychophysiological interaction (PPI) analysis (*Friston et al., 1997*). The IPL and MFG regions underneath the tACS electrodes were used as seeds, allowing us to test whether their interactions were influenced in a phase dependent manner. Synchronous tACS during the 2-back task increased FC between the right IPL and posterior parts of the left dorsolateral prefrontal cortex (DLPFC) (*Figure 5* and *Supplementary file 1*, table supplement 6) compared to the FC observed during task performance in the absence of stimulation. In contrast, no modulation of FC was observed for the MFG-electrode seed. Desynchronous tACS produced increases in FC in occipito/temporal regions in a

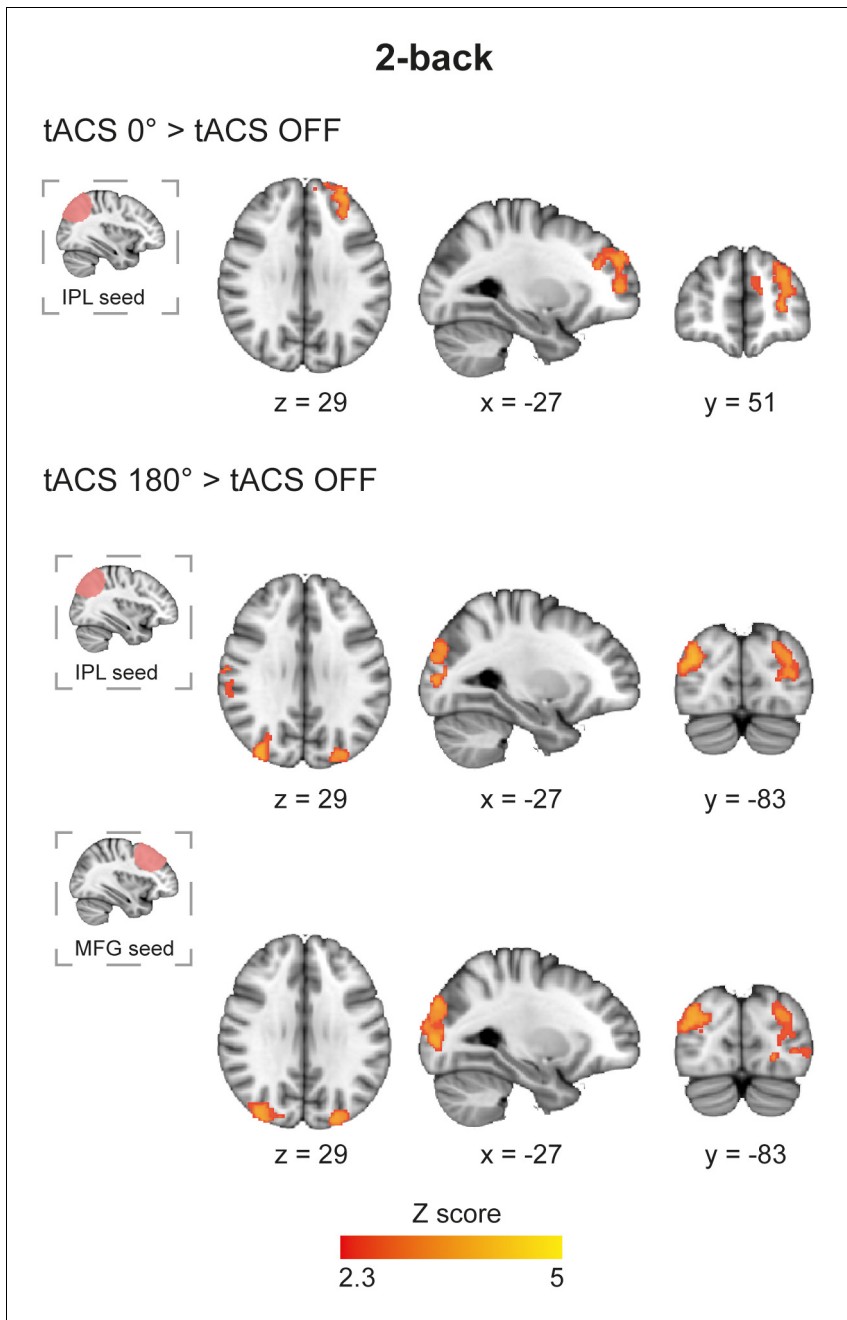

**Figure 5.** Functional connectivity in the 2-back task during synchronous and desynchronous tACS. Differential connectivity for synchronous versus desynchronous tACS during the 2-back task (n = 21). Synchronous tACS resulted in increased functional connectivity in comparison to tACS OFF between the parietal electrode region, IPL-electrode, and the dorsolateral prefrontal cortex during task performance. The desynchronous tACS condition increased functional connectivity between the parietal and frontal stimulated regions (IPL-electrode and MFG-electrode, respectively) and occipito-temporal regions. Images are in the Montreal Neurological Institute (MNI) space coordinates and in radiological space. All images have been thresholded with FSL clusterwise correction $Z > 2.3$, $p < 0.05$.

The following source data is available for figure 5:

**Source data 1.** This folder contains the MRI contrast maps in *Figure 5*, both thresholded (that is, corrected for multiple comparison using cluster correction) and non-thresholded.

similar pattern for both MFG- and IPL-electrode seeds (*Figure 5* and *Supplementary file 1*, table supplement 6). There was no correlation between FC in the 2-back task and RTs. The tACS phase dependent modulation of FC observed for the 2-back task was absent for the CRT task, where no modulations of FC were observed. As a control, we also investigated functional connectivity using the return electrode region as seed, which did not show any modulations in FC in either the 2-back or CRT tasks. Furthermore, no modulation of FC was observed during fixation periods using any of the seed regions for any tACS or task conditions.

## Control analyses

We assessed whether our results might be confounded by motion, heart rate or the subjective perception of different tACS conditions. Our fMRI data were carefully cleaned to remove motion and structured artifacts (for example, scanner artifacts, non-neural physiological noise, etc.) by removing noise components using independent component analyses (*Griffanti et al., 2014*; *Salimi-Khorshidi et al., 2014*). Overall, there was very low movement in our sample (mean framewise displacement (FD) for 2-back = 0.044 ± 0.013, n = 21, and CRT = 0.040 ± 0.011, n = 20) and there was no difference in FD between periods with and without stimulation or for different tACS phase conditions for any of the tasks (Repeated measures ANOVA for CRT and 2-back with factors stimulation (ON, OFF) and tACS phase (0°, 180°); all F values < 2, all p values > 0.2).

Similarly, heart rate (available for a subset of participants; N = 6) was not different between periods with and without stimulation or between different tACS phase conditions (Repeated measures ANOVA for CRT and 2-back with factors stimulation (ON, OFF) and tACS phase (0°, 180°); all F values < 2, all p values > 0.4). In addition, analysis of post-experiment questionnaires showed that participants were unaware whether or not they were receiving stimulation and could not distinguish between the type of stimulation (Materials and methods).

## Discussion

Our results demonstrate that external manipulations of oscillatory synchrony across a right-sided frontoparietal network impact neural activity and modify behavior. The effects depend on the relative phase of the external oscillators and also the underlying cognitive state at the time tACS is applied. In particular, we showed that imposing theta frequency synchronously across the inferior parietal lobe and middle frontal gyrus improves verbal WM performance and increases frontal and parietal brain activity. The functional connectivity of the inferior parietal lobe was also modulated by tACS in a phase dependent manner. Frontal interactions increased with synchronous theta stimulation whereas posterior interactions increased with desynchronous stimulation, suggesting that information flow through brain regions involved in WM can be differentially affected by varying the phase of external stimulation. Therefore, we provide a direct demonstration of the role of phase synchronization in cognitive demanding WM processes and we showed the neural correlates associated with this process.

It has been proposed that tACS interacts with ongoing oscillations and the resultant neural modulations spread along brain networks (*Fröhlich and McCormick, 2010*; *Ali et al., 2013*), with entrainment being more effective when the induced frequency matches the endogenous rhythm (*Fröhlich and McCormick, 2010*; *Reato et al., 2010*). This leads to the prediction that stimulation is highly dependent on the underlying network dynamics (*Alagapan et al., 2016*). Our imaging results for the 2-back and CRT tasks strongly support these notions and show how the physiological effect of external oscillatory manipulations are highly dependent on the underlying cognitive state. While synchronous and desynchronous stimulation produced distinct activity and connectivity patterns in the 2-back task, they resulted in similar changes of activity and no effects on connectivity for the CRT task. Compared to the CRT, the 2-back task is more demanding and engages the frontoparietal network more heavily, which is observed both in terms of increased BOLD activity and theta power (*Payne and Kounios, 2009*; *Heinzel et al., 2014*)

Our behavioral findings strongly suggest a causal link between theta phase coupling in the frontoparietal network and cognition. We replicate the observation that synchronous tACS applied across the frontoparietal networks improves WM performance (*Polanía et al., 2012*), and extend these results by showing that this cognitive enhancement is dependent on the demands of the cognitive task. Our observations that tACS had no effect on less demanding 1-back or CRT tasks suggests

that the behavioral effects of entraining oscillations interacts with the extent that a network is already engaged by a cognitive task. Remarkably, synchronous tACS produced sufficient improvement in response times to make 2-back performance similar to the 1-back condition, which is typically associated with much faster responses.

Increased rhythmic synchrony across a network is thought to improve information processing by increasing network efficiency, an effect particularly important during demanding cognitive processing (*Fries, 2005*, *2009*; *Deco et al., 2011*). For example, theta synchrony between frontal and parietal brain regions increases during complex manipulations of items held in WM (*Sauseng et al., 2005*). Importantly, our neuroimaging results show the neural correlates of this effect on network efficiency. Entraining synchronous activity across the right frontoparietal network was associated with increased activity within the inferior parietal lobe and middle frontal gyrus, regions known to play a key role in WM function. Activation in these regions increases with WM demands and its damage leads to impairments of WM (*Curtis and D'Esposito, 2004*; *Berryhill and Olson, 2008*; *Fedorenko et al., 2013*). In keeping with these findings, we show that increased activity within the right inferior parietal lobe correlated with faster responses during the 2-back, supporting a link between neural activity and the behavioral effects of tACS. This is in agreement with the critical role of this region during WM maintenance (*Constantinidis and Steinmetz, 1996*; *Pessoa et al., 2002*), and the observations that disrupting its activity through transcranial magnetic stimulation affects performance in both spatial and verbal WM tasks (*Kessels et al., 2000*; *Mottaghy et al., 2003*; *Postle et al., 2006*). Furthermore, during maintenance of items in WM the parietal cortex is positively associated with trial-by-trial performance and inter-individual differences in WM capacity (*Todd and Marois, 2004*; *Xu and Chun, 2006*). The fact that a relationship with response times was observed for the parietal but not the frontal region might be explained by the different roles attributed to parietal and frontal cortices in verbal WM. Evidence suggests that these regions have different contributions to phonological storage and executive control (*Paulesu et al., 1993*). Moreover, our results are congruent with findings from other studies showing correlations between BOLD activity or connectivity density and response times in similar verbal WM tasks (*Honey et al., 2000*; *Tomasi et al., 2011*; *Liu et al., 2017*). Thus, by increasing neural activity in the parietal region, tACS might have interacted with the physiological mechanisms associated with response production.

Importantly, the changes in BOLD we observed were not constrained to regions close to the stimulation sites. Rather a more general effect across salience, visuospatial and basal ganglia networks was seen, which are all regions involved in WM performance (*Pessoa et al., 2002*; *Owen et al., 2005*). Interestingly, the stimulant drug methylphenidate has been shown to enhance WM performance and increase parietal activity in a similar verbal N-back task (*Tomasi et al., 2011*), suggesting that cognitive enhancement produced pharmacologically or electrically may converge on similar neural mechanisms.

As the maintenance of information in WM relies on the coordination of distant brain regions, the effects of tACS on functional connectivity may inform the mechanisms by which oscillations modulate long-range connectivity. During synchronous stimulation, functional connectivity increased between the IPL-electrode region and frontal parts of the frontoparietal network, that is, DLPFC. Increased frontoparietal functional connectivity has previously been observed during WM processing (*Fell and Axmacher, 2011*), is associated with higher accuracy and faster RTs (*Prado et al., 2011*), and is linked to increased theta synchrony (*Liebe et al., 2012*). Our results extend these findings by showing how externally induced theta synchrony can increase frontoparietal interactions between the parietal cortex and DLPFC that are important for WM processing. This pattern was observed when the parietal region was used as a seed, but not for the frontal region. One possible explanation for this finding is that functional connectivity between the frontal and other brain regions was already operating at peak levels in the task periods without stimulation and could not be further modulated by tACS. A more intriguing possibility is that distinct frequency channels are responsible for carrying feedforward and feedback signalling. Such a distinction has been observed in the visual cortex, where feedforward influences are carried by theta and gamma band synchronization, while feedback influences by beta band synchronization (*Bastos et al., 2015*). In this framework, it is possible that tACS applied at the theta frequency would differentially enhance feedforward connectivity across the synchronized fronto-parietal network. Studies combining tACS with electrophysiological methods could help explore this hypothesis.

While synchronous stimulation resulted in increased functional connectivity between parietal and frontal areas, desynchronous tACS was associated with increased connectivity to occipital regions, which could suggest an increase in information exchange between higher- and low-level areas. The shift in parietal functional connectivity between frontal and posterior regions shows how altering phase synchrony might control long-range network interactions and so shape information flow. One interpretation of our findings is that high WM demands are normally associated with high levels of frontoparietal connectivity, and this can be enhanced by synchronous tACS applied across the network. Disrupting interactions across the network through desynchronous tACS increases parietal connections to the occipital lobe, perhaps as a result of the restoration of a 'default' pattern of functional connectivity normally observed in the absence of high levels of top-down cognitive control. This idea is supported by the notion that phase relations among neuronal groups could contribute to selective routing of information and shape effective connectivity, an argument supported by computational models (*Akam and Kullmann, 2014*) and empirical evidence (*Womelsdorf et al., 2007*; *Helfrich et al., 2014*). Intracortical recordings show that altered phase relations between brain regions precede changes in neural activity, providing evidence that the influence of synchronizing activity across neuronal groups can be phase dependent (*Womelsdorf et al., 2007*). Converging evidence that this can be externally modulated is provided by the observation that in- and out-of-phase tACS applied in the gamma band modulates inter-hemispheric connectivity and shapes visual perception (*Helfrich et al., 2014*).

Our study has a number of limitations. First, there is unbalanced electric field distribution between the tACS conditions (see *Figure 1—figure supplement 1*), which is a consequence of using a common return electrode. Thus, when current was applied synchronously to the frontal and parietal electrodes the temporal return electrode received the sum of the applied currents to each electrode, while in the desynchronous condition the current in the return electrode is cancelled by the opposing phases of the frontal and parietal electrodes. Nevertheless, although synchronous and desynchronous tACS resulted in different current distributions in the brain, our results could not have been predicted by a current density model. Modulations of brain activity and connectivity were predominantly restricted to areas involved in task performance and no effects were observed in the cortical area underneath the temporal return electrode. These results further demonstrate that the effects of brain stimulation cannot be determined without taking into account the underlying brain dynamics (*Reato et al., 2013*) and provide additional support for the critical neural state-dependency of tACS (*Feurra et al., 2013*; *Ruhnau et al., 2016*). Second, the area of stimulation was relatively large and potentially affected subregions with complex functional architecture and diverse effects on WM processing. Future work will need to use high-density multi-channel stimulators to improve the focality of the cortical effect of tACS. Third, the optimal frequency and phase is likely to suffer from inter-individual variability. However, the parameter space of possible stimulation regimes is very large and it is likely that other combinations of tACS parameters (that is, frequency, phase, intensity, etc.) will produce greater effects on cognition. The key to unlocking the potential of this technique for clinical use will be to understand the neural mechanisms governing the cognitive effects of the stimulation. Machine-learning techniques combined with real-time imaging could help select the ideal combination of parameters to induce the desired modulation of brain activity/connectivity in a subject-specific manner (*Lorenz et al., 2016*). Finally, while our fMRI approach benefited from good spatial resolution in the absence of complex electrical artifacts, we are not able to specify which brain frequencies were modulated by our intervention, as the correspondence between brain oscillations and BOLD signal is not fully understood (*Scheeringa et al., 2011*) and tACS might induce cross-frequency coupling (*Reato et al., 2010*).

Other studies have shown that additional forms of non-invasive transcranial electrical stimulation (tES), particularly transcranial direct current stimulation (tDCS) modulate WM performance. The majority of these studies targeted the DLPFC and recent meta-analyses have indicated small but significant effects of tDCS on WM (*Brunoni and Vanderhasselt, 2014*; *Hill et al., 2016*; *Mancuso et al., 2016*). Broadly, these findings and ours show that tES is capable of modulating the neural processes associated with WM. But are tDCS and tACS acting through similar mechanisms? Although tDCS uses direct instead of an oscillatory current to modulate cortical excitability, anodal tDCS applied to the DLPFC increased oscillatory brain activity in alpha and theta frequency bands in occipito-parietal regions during a verbal WM task (*Zaehle et al., 2011*). This indicates that the local changes in neuronal excitability induced by tDCS produced interactive effects that resulted in the

modulation of oscillatory activity in distant cortical regions. Although this shows that tDCS interacts with the neural mechanisms associated with WM, future studies should carefully consider the biological processes they aim to target, as different tES modalities impact brain excitability to different degrees (*Inukai et al., 2016*). Furthermore, a recent study elegantly demonstrated that WM performance is very sensitive to the external stimulation parameters (*Alekseichuk et al., 2016*). A study comparing different tES modalities could help answer this question. Such a study would benefit from applying a similar methodology to the one we employed, in which blocks of short durations of different tES modalities could be combined with fMRI.

Overall, our findings indicate a direct link between behavioral performance in a demanding working-memory task and large-scale brain synchronization across a right frontoparietal network activated by the task. We showed how manipulations of tACS phase modulate the underlying brain activity and that tACS can influence long-range connectivity in a phase- and brain state- dependent manner. More generally, our work shows the potential of performing simultaneous tACS-fMRI to understand the neural mechanisms underlying externally induced oscillatory synchronization.

## Materials and methods

### Participants

Experiment 1 included 10 healthy volunteers (six females, age range: 21–40 years, mean age ± SD: 28.6 ± 5.0), all right-handed. In Experiment 2, 24 healthy volunteers (21 new participants, nine females, age range: 20–35 years, mean age ± SD: 27.2 ± 4.4), 21 right-handed participated in the study. Two participants were excluded from Experiment 2, one due to excessive movement in the scanner and another due to low accuracy in the tasks. Additionally, two other participants were excluded from the CRT analysis in Experiment 2, one due to unreliable reaction times (>20% of the trials had RTs > 1.5 SD above the mean for the subject individual RT) and another due to a technical error with the recordings of the behavioral fMRI responses. One participant was excluded from the 2-back analysis in Experiment 2 due to low accuracy (performance at chance level). Thus, for Experiment 2 our final cohort for the CRT task was composed of twenty subjects, three left-handed, seven females, age range: 20–35 years, mean age ± SD: 26.96 ± 4.34, and twenty-one subjects, three left-handed, seven females, age range: 20–35 years, mean age ± SD: 27.38 ± 4.56 for the 2-back task.

All subjects were educated to degree level or above, with no history of neurological or psychiatric illness. Participants gave written informed consent. The study conforms to the Declaration of Helsinki and ethical approval was granted through the local ethics board (NRES Committee London – West London and GTAC).

The sample size in Experiment 1 was chosen based on the effect sizes, between synchronous and desynchronous stimulation, observed in a previous study showing phase-dependent tACS modulations of RTs in a WM task (*Polanía et al., 2012*). In Experiment 2, the sample size was increased by approximately 50% to allow greater across-subject variability in BOLD related variables.

### Tasks

Participants performed tasks with different levels of cognitive difficulty. In Experiment 1 participants performed two tasks, the choice-reaction task (CRT) and two versions of the N-back task: 1-back and 2-back. In Experiment 2 participants performed the CRT and the 2-back tasks in separate runs in a block design paradigm, with alternating periods of task (30 s) and fixation at a cross (20 s).

The CRT is a 2-alternative force choice paradigm used for testing general alertness and motor speed. Participants were instructed to respond as quickly and as accurately as possible, with the right or left index finger, to the presentation of a right or left pointing arrow (size = 3 degrees of visual angle horizontally and 2.26 vertically) (*Figure 1A*). Each trial started with a fixation cross presented centrally (size = 1 × 1 degrees of visual angle) for 350 ms, followed by a left or right arrow (1200 ms) and then by a blank period of 450 ms. Participants performed 180 trials.

The N-back is a WM task that requires on-line monitoring, and manipulation of remembered information. Participants were presented with a single digit number (0–9) sequentially (*Figure 1B*). Each digit (Arial font style, height was 2 degrees of visual angle) was displayed for 500 ms followed by a fixation (size = 1 × 1 degrees of visual angle) cross for 1500 ms. In the 2-back task participants detected a repetition of the digit occurring two trials before, whereas in the 1-back task they

detected whether the stimulus matched the one presented in the previous trial. Target trials occurred in 25% of the trials. Participants performed 270 trials in Experiment 1 and 180 trials in Experiment 2. Subjects were instructed to respond as quickly and as accurately as possible using their right index finger.

Stimuli were designed using the Psychtoolbox (*Brainard, 1997*) for Matlab (Mathworks, Natick, MA). In Experiment 1 responses were collected through a custom-made response box and in Experiment 2 through a fiberoptic response box (NordicNeuroLab, Norway), interfaced with the stimulus presentation PC.

## Transcranial alternating current stimulation (tACS)

Stimulation was delivered using two MR-compatible battery-driven stimulators (NeuroConn GmbH, Ilmenau, Germany). The stimulation electrodes (5 cm diameter 'donut' shaped rubber electrodes) were positioned with their center locations at F4 and P4 and the return electrode at T8, according to the International 10–20 system. Impedances were kept below 10 kΩ using a conductive paste (Ten20, D.O. Weaver, Aurora, CO, USA), which also held the electrodes in place. Stimulation was sinusoidal, 6 Hz frequency, peak-to-peak amplitude of 1000 µA, and no DC offset.

TACS was delivered in two different conditions: (1) *Synchronous condition (0°)* – Electrodes F4 and P4 with a 0° phase offset and return electrode at T8; (2) *Desynchronous condition (180°)* – same montage as in the 'synchronous' condition, but electrodes F4 and P4 had a 180° phase offset. In both conditions stimulation began with a 1 s ramp-up and ended with a 1s ramp-down. In the desynchronous condition the current in one channel was kept constant at 1000 µA after ramp-up for half a cycle.

In Experiment 1, participants performed one session of each tACS condition and sham in a pseudo-randomized cross-over design. Each condition was run in separated sessions at least one day apart (range 1–8 days, mean 2.5 ± 1.7). Stimulation was administered for the duration of the tasks, starting at the beginning of each task and finishing when the task was over, total stimulation duration was 26.5 min. During sham sessions, tACS was applied in the synchronous condition for 30 s at the beginning of each task. In Experiment 2, tACS was applied in a pseudo-randomized order for the duration of task (30s) or fixation blocks (20s), where a fixation cross was presented centrally. The application of short trains of tACS has the advantage of allowing the measurement of many periods of neural entrainment without an increase of oscillatory power due to tACS aftereffects (*Vossen et al., 2015*). Each participant performed one fMRI run of each task and tACS condition (that is, 2-back synchronous tACS, 2-back desynchronous tACS, CRT synchronous tACS and CRT desynchronous tACS). The order of the runs was counterbalanced across participants. In each run participants performed 24 blocks (that is, six blocks/condition: 'task + tACS ON', 'task + tACS OFF', 'fixation + tACS ON', 'fixation + tACS OFF'- see *Figure 2B*). Total stimulation duration in Experiment 2 was 20 min (11 min 36 s per run). A subset of participants (n = 13) were questioned after each block whether they perceived the tACS stimulation; the question 'Have you had stimulation?" appeared on the screen for 4 s accompanied by the words 'Yes' in the lower right corner and 'No' in the lower left corner; participants were requested to press the corresponding button (right for 'Yes' and left for 'No'). Experiment 2 was optimized to detect neurophysiological changes due to tACS by increasing signal-to-noise ratio and allowing computation of psycho-physiological interaction analyses.

Stimulators were controlled and monitored through Spike2 software via a Micro1401–3 data acquisition unit (both Cambridge Electronic Design, Cambridge, UK). The beginning and end of each stimulation block was controlled via an external trigger sent to the Micro1401–3 from the computer running the experimental paradigm (which received TTL triggers from the MR scanner), this ensured that task and stimulation were synchronous to the scanner clock (Experiment 2). The experimental setup for Experiment 2 is represented in *Figure 2A*. In brief, the stimulators were placed outside the MR shielded room and the current from the stimulators was delivered into the scanner room after being filtered from RF noise by two filter boxes, one placed in the operator room and another inside the scanner bore, the latter was connected to the stimulation electrodes via a Y-cable. Phantom and pilot experiments were initially conducted to ensure that the experimental setup did not introduce artifacts in the fMRI signal. In agreement with previous studies (*Antal et al., 2014*; *Cabral-Calderin et al., 2016*) the tACS montage employed in the study did not cause image artifacts.

For each subject that took part in Experiment 2 we computed the electric field distributions on the cortex resulting from the two tACS conditions using a realistic finite element model as implemented in SimNibs 2.0 (*Opitz et al., 2015*; *Thielscher et al., 2015*). Briefly, different tissue types including white matter, grey matter, CSF, skin and skull were segmented and tetrahedral volume meshes of the head were generated using the SimNibs routine. In the case of the synchronous tACS condition three electrodes were included in the model: two anodes (F4 and P4) and one cathode (T8). In the case of the desynchronous tACS condition the current in the cathode is cancelled by the opposing flow of currents in the anodes; this condition was modeled with an anode and a cathode (P4 and F4). The localization of each electrode was defined by transforming the electrode center coordinates in MNI space to the space of the individual meshes (native FreeSurfer (http://surfer.nmr.mgh.harvard.edu/) RAS space). The electric field distributions were very similar across participants (*Figure 1—figure supplement 1*).

## tACS questionnaire

Before each experiment all participants were exposed to tACS stimulation for a short period of time (<1 min) to ensure that they were comfortable to proceed with the experiment. Participants were blind to the stimulation condition applied. At the end of each experiment, participants completed a questionnaire to assess possible side-effects of the tACS stimulation by rating from 0 (none) to 4 (severe) the intensity and duration of: pain, burning, warmth/heat, itchiness, pitching, metallic taste, fatigue, skin flush, effect on performance or any other side-effect perceived.

Analysis of the tACS questionnaires for Experiment 1 indicated that participants were unaware whether or not they received stimulation (50% of the participants reported perceiving stimulation in the synchronous condition, 70% in desynchronous condition and 70% in the sham condition). In Experiment 2, participants' demonstrated low accuracy for correctly detecting stimulation blocks (2-back task: tACS 0° = 34.52% ± 34.26; tACS 180° = 17.86% ± 26.52; CRT task: tACS 0° = 41.03% ± 27.73; tACS 180° = 5.13% ± 21.93), which were explained by participants either perceiving stimulation during non-stimulation periods, or not perceiving the stimulation at all. The latter could possibly be explained by habituation due to frequent short trains of stimulation. Importantly, there was no difference between the accuracy for perceiving stimulation between the synchronous and desynchronous tACS conditions in the 2-back task ($t_{(13)}$ = 1.59, p=0.136), indicating that our results were not influenced by participants perceiving the two tACS conditions differently.

There was no significant difference between tACS conditions (synchronous, desynchronous and sham) for any of the side effects (Chi Square tests), for Experiment 1. There was no difference in side effects between Experiment 1 and Experiment 2. None of the participants reported perceiving phosphenes.

## Pulse oximetry recordings

The pulse oximetry signal was acquired concurrently with fMRI using the integrated Siemens Physiological Monitoring Unit. Recordings were made using a photoplethysmograph (5 ms sampling rate) with an infrared emitter placed under the pad of the right little finger. Registration to the fMRI images was made using a time stamp from the scanner's output file. Due to technical difficulties we only obtained time stamps for a subset of participants (N = 8 initially and N = 6 after exclusion criteria, see 'Participants' paragraph above) and therefore only those were analysed. Signal analysis was performed using in house scripts written in Matlab (R2013b, MathWorks).

## Image acquisition

Scanning was performed on a 3T Siemens Verio (Siemens, Erlangen, Germany), using a 32-channel head coil. FMRI images were obtained using a T2*-weighted gradient-echo, echoplanar imaging (EPI) sequence, 3 mm$^3$ isotropic voxel, repetition time (TR) 2 s, echo time (TE) 30 ms, flip angle (FA) 80°, field of view 192 × 192 × 105 mm, 35 slices, GRAPPA acceleration factor = 2, 348 volumes. Standard T1-weighted structural images were acquired using an MP-RAGE sequence, 1 mm$^3$ isotropic voxel, TR 2.3 s, TE 2.98 ms, inversion time 900 ms, FA 9°, field of view 256 × 256 mm, 256 × 256 matrix, 160 slices, GRAPPA acceleration factor = 2.

## Image analysis

Data preprocessing was performed using the FMRI Expert Analysis Tool (FEAT) Version 6.00, from the FMRIB's Software Library (FSL [*Smith, 2004*; *Jenkinson et al., 2012*]). We performed motion correction using MCFLIRT (*Jenkinson et al., 2002*), removal of low-frequency drifts (high-pass filter of 0.01 Hz), spatial smoothing (Gaussian kernel filter with a full width at half maximum of 6 mm), brain extraction to remove non-brain tissue (BET [*Smith, 2002*]), and coregistration using FMRIB's Nonlinear Image Registration tool (FNIRT) to register the participant's fMRI volumes to Montreal Neurological Institute (MNI) 152 standard space using the T1-weighted scan as an intermediate.

Single-session ICA was performed for each run using Multivariate Exploratory Linear Optimized Decomposition (MELODIC [*Beckmann et al., 2005*]). The resulting components were automatically classified into signal and noise using FMRIB's ICA-based Xnoiseifier (FIX [*Griffanti et al., 2014*; *Salimi-Khorshidi et al., 2014*]). FIX was previously trained in an independent cohort of twenty individuals acquired in the same scanner with the same imaging parameters. Classifications were manually inspected and adjusted when required. Independent components classified as noise components were subsequently removed from each voxel's time series.

### Univariate fMRI analyses

Subject-level general linear models (GLM) included regressors of interest for task blocks with and without tACS and fixation blocks with tACS (resulting in three regressors: 'task + tACS ON', 'task + tACS OFF', 'fixation + tACS ON'; implicit baseline consisted of 'fixation + tACS OFF'). Regressors were created by convolving a boxcar kernel with a canonical double-gamma hemodynamic response function. The GLM design matrix consisted of those regressors of interest, their first temporal derivatives and six movement regressors to account for movement-related noise, plus a nuisance regressor for stimulation perception (for subjects where this was assessed). Mixed effects analyses of group effects were performed for each task separately using FLAME 1, the FMRIB local analysis of Mixed Effects (*Beckmann et al., 2003*; *Woolrich et al., 2004*), for the regressors of interest and the contrasts 'task + tACS ON' vs 'task + tACS OFF'. Because the two tACS phase conditions were delivered in separate runs a third-level two-sample paired t-test was performed for the contrasts ('task + tACS ON 0°" > 'task + tACS OFF') > ('task + tACS ON 180°" > 'task + tACS OFF') and the inverse contrast. The final Z statistical images were thresholded using Gaussian Random Fields based cluster inference with an initial cluster-forming threshold of $Z > 2.3$ and a family-wise error (FWE) corrected cluster-extent threshold of $p<0.05$.

### TACS ROIs

Masks corresponding to the position of the tACS electrodes were generated using MANGO (Multi-image Analysis GUI) software (http://ric.uthscsa.edu/mango/mango.html) by defining 25 mm radius spheres centered on the electrodes' centre coordinates. These were obtained from the projection of the electrode position onto the cortical surface (*Koessler et al., 2009*) and converted to MNI space using the Nonlinear Yale MNI to Talairach Conversion Algorithm (*Lacadie et al., 2008*) (F4: x = 42, y = 24, z = 46; P4: x = 43, y=−69, z = 47; T8: x = 71, y=-16, z=−10). Areas outside the cortex were removed and the masks were registered into each participant's functional images in MNI space.

### Functional Connectivity analysis

Whole brain psychophysiological interaction (PPI) analyses (*Friston et al., 1997*) were performed to explore whether tACS modulates brain connectivity. We used as seed regions the masks for P4 and F4 tACS electrodes. Time-courses for the seed regions were extracted for each participant (physiological term). Separate PPI models were created for each task and each tACS stimulation condition. Regressors included: (1) all task periods (task), to explain the variance associated with the task; (2) blocks of tACS stimulation during task; (3) blocks of tACS stimulation during fixation; (4) six movement regressors to account for movement-related noise; and (5) a nuisance regressor for stimulation perception (for subjects where this was assessed). Models were created with psychological terms for tACS stimulation during task, tACS stimulation during fixation and all task periods. For each model, the physiological term and the psychological term were used to create the PPI interaction term, the remaining regressors were also included in the model. The first-level PPIs were then entered into a group-level regression analysis for each model and tested with t-contrasts 1 or −1 to reveal positive

or negative PPI effects, respectively. Higher-level analysis were performed for the contrasts: tACS 0° > tACS OFF, tACS 180° > tACS OFF in both directions. The final Z statistical images were thresholded using Gaussian Random Fields based cluster inference with an initial cluster-forming threshold of $Z > 2.3$ and a family-wise error (FWE) corrected cluster-extent threshold of $p<0.05$.

## Statistical analysis

Data analyses were performed using SPSS (PASW Statistics Release 22.0.0) and Matlab (R2013b, MathWorks). For the behavioural tasks we calculated mean reaction times (RT) and accuracy (defined as the percentage of correct responses). Only correct RTs were included in the analysis. RTs were normally distributed (Shapiro–Wilk test) for all subjects and all conditions, with the exception of one subject for the CRT task in the sham condition. Data from this subject was included in the analysis and excluding this subject did not change the results. Analyses of behavioural data were performed using repeated-measures ANOVA and missing data handled using listwise deletion. Thus, data of nineteen participants were included for the behavioural analysis of Experiment 2. Effect sizes were measured by calculating the partial eta squared ($\eta_p^2$) for repeated-measures ANOVAs and Cohen's d for post hoc t-test comparisons.

## Acknowledgements

We thank Thomas Reed for helping in data acquisition and Dr. Jonathan Howard for methodological and technical assistance. We are grateful to Prof. Michael Hausser for critical reading of the manuscript and Dr. Sara di Simoni for proofreading.

## Additional information

### Funding

| Funder | Grant reference number | Author |
| --- | --- | --- |
| Wellcome | 103045/Z/13/Z | Ines R Violante |
| Wellcome | 103429/Z/13/Z | Lucia M Li |

The funders had no role in study design, data collection and interpretation, or the decision to submit the work for publication.

### Author contributions

IRV, Conceptualization, Data curation, Formal analysis, Funding acquisition, Investigation, Methodology, Writing—original draft, Project administration, Writing—review and editing; LML, Conceptualization, Data curation, Formal analysis, Investigation, Writing—review and editing; DWC, RLe, Validation, Methodology, Writing—review and editing; RLo, Methodology, Writing—review and editing; AH, Conceptualization, Writing—review and editing; JCR, Conceptualization, Funding acquisition, Writing—review and editing; DJS, Conceptualization, Funding acquisition, Project administration, Writing—review and editing

### Author ORCIDs

Ines R Violante, http://orcid.org/0000-0002-4787-2901
Robert Leech, http://orcid.org/0000-0002-5801-6318

### Ethics

Human subjects: The study conforms to the Declaration of Helsinki and ethical approval was granted through the local ethics board (NRES Committee London - West London & GTAC). All subjects were educated to degree level or above, with no history of neurological or psychiatric illness. Participants gave written informed consent.

## Additional files

**Supplementary files**
• Supplementary file 1. Summary tables of the results from fMRI whole-brain analysis for the statistical maps shown in *Figure 4* and *Figure 5*. For each cluster, the peak and five local maxima within the cluster are listed along with x-y-z locations in MNI space. R = right hemisphere; L = left hemisphere.

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
