## [Decision Letter]

Thank you for submitting your article "Externally induced frontoparietal synchronization modulates network dynamics and enhances working memory performance" for consideration by *eLife*. Your article has been reviewed by two peer reviewers, and the evaluation has been overseen by a Reviewing Editor and Sabine Kastner as the Senior Editor. The reviewers have opted to remain anonymous.

The reviewers have discussed the reviews with one another and the Reviewing Editor has drafted this decision to help you prepare a revised submission.

Summary:

In this manuscript, the authors report the results of two well-thought out and well-described experiments that provide convincing evidence that synchronous rhythmic activity in a large-scale distributed frontoparietal network is a critical mechanism mediating working memory. Prior evidence has suggested that oscillatory activity in these brain regions occurring the theta range (4-8 Hz) is associated with working memory tasks, and that the relative phases of these oscillations impact performance in this domain. Extending prior work in this area, the investigators applied theta (6 Hz) transcranial alternating current stimulation (tACS)-a technology that can exogenously manipulate neuronal oscillatory activity-to right frontal and parietal regions thought to support working memory. In their first experiment, they found that stimulation enhanced working memory performance, specifically when subjects were engaged in a task in which working memory demands were high (in this case, a 2-back N-back task). In a second experiment, the investigators applied tACS in conjunction with functional magnetic resonance imaging (fMRI), and found that synchronous tACS applied during the working memory task increased parietal activity, which in turn correlated with enhanced performance. Moreover, the investigators found that the phase of tACS (synchronous= 00 relative phase; asynchronous= 1800 relative phase) directly influenced functional connectivity between within the presumed right frontoparietal working memory network. Reviewers were generally enthusiastic about this manuscript, and provided comments that were both constructive in nature and readily addressable.

Essential revisions:

1) In the Introduction, the overarching term 'WM' is used, but only in one place in the Introduction is it noted that verbal WM is tested. It is important to clarify throughout the manuscript that verbal WM is the focus of the paper. A more comprehensive claim about WM in general would require data from multiple types of stimuli.

2) Methods/Results: In the Experiment 2 behavioral analyses, it is not entirely clear how the authors dealt with the fact that an unequal number of different subjects performed each of the two tasks – especially since it seems that the factor "Task" was treated as a within-subjects variable.

3) Results: Explain Experiment 2 behavioral results in more detail, specifically to indicate in which direction the changed RT's were linked with tACS condition. Include effect sizes.

4) In the Figure 1—figure supplement (finite element modeling), please address why superior temporal regions seem to be the most strongly affected here.

5) Discussion: The authors should provide an explanation as to why the tACS effects were somewhat different in terms of BOLD activity and FC at the two electrode regions (parietal vs. frontal). In particular, why does BOLD activation at the parietal, but not frontal, electrode region correlate with performance? Similarly, why would increased FC within the frontoparietal network during synchronous tACS only be present when the parietal, but not frontal, electrode region is used as the seed? The reason for these differences is not obvious, and therefore should be addressed.

6) Discussion: Would the authors care to speculate how findings from forms of non-invasive stimulation, e.g., tDCS, relate to the current data? It seems as if this other literature would be relevant to the current Discussion and that it would add breadth to the interested audience.

---

## [Author Response]

Essential revisions:

1) In the Introduction, the overarching term 'WM' is used, but only in one place in the Introduction is it noted that verbal WM is tested. It is important to clarify throughout the manuscript that verbal WM is the focus of the paper. A more comprehensive claim about WM in general would require data from multiple types of stimuli.

We have now revised the manuscript to clarify that verbal WM was tested.

2) Methods/Results: In the Experiment 2 behavioral analyses, it is not entirely clear how the authors dealt with the fact that an unequal number of different subjects performed each of the two tasks – especially since it seems that the factor "Task" was treated as a within-subjects variable.

The behavioral analysis of Experiment 2 was performed using repeated measures ANOVA. Indeed, due to exclusion criteria there were an unequal number of subjects per task, i.e. twenty-one for the 2-back and twenty for the CRT task. The method employed to deal with missing cases was listwise deletion, meaning that the analysis was performed only including the cases in which all variables were present. Thus, behavioral analysis for Experiment 2 was performed on nineteen subjects for whom both CRT and 2-back data was present. This is now specified in the Methods section (“Statistical analysis”) of the manuscript":

“Analyses of behavioral data were performed using repeated-measures ANOVA and missing data handled using listwise deletion. Thus, data of nineteen participants was included for the behavioral analysis of Experiment 2.”

3) Results: Explain Experiment 2 behavioral results in more detail, specifically to indicate in which direction the changed RT's were linked with tACS condition. Include effect sizes.

We have now included effect sizes for both Experiment 1 and 2 in the Results section. Effect sizes were measured by calculating the partial eta squared (η_p_^2^) for repeated-measures ANOVAs and Cohen’s d for post hoc t-test comparisons. This description was also included in the Methods section.

4) In the Figure 1—figure supplement (finite element modeling), please address why superior temporal regions seem to be the most strongly affected here.

This is a consequence of the electrode montage used in the study which caused a unbalanced electric field distribution between the tACS conditions. Similarly to previous studies (Polaniaet al., Curr Biol 2012; 22(14): 1314-8; Polaniaet al., Nat Commun 2015; 6:8090), our montage included a common return electrode. This means that when current was applied synchronously to the frontal and parietal electrodes the temporal return electrode received the sum of the applied currents to each electrode, while in the desynchronous condition the current in the return electrode is cancelled by the opposing phases of the frontal and parietal electrodes. We have now addressed this more clearly in the Discussion section:

“…there is an unbalanced electric field distribution between the tACS conditions (see figure supplement 1). This is a consequence of using a common return electrode. […] These results further demonstrate that the effects of brain stimulation cannot be determined without taking into account the underlying brain dynamics (Reato et al., 2013) and provide additional support for the critical neural state-dependency of tACS (Feurra et al., 2013, Ruhnau et al., 2016).”

5) Discussion: The authors should provide an explanation as to why the tACS effects were somewhat different in terms of BOLD activity and FC at the two electrode regions (parietal vs. frontal). In particular, why does BOLD activation at the parietal, but not frontal, electrode region correlate with performance? Similarly, why would increased FC within the frontoparietal network during synchronous tACS only be present when the parietal, but not frontal, electrode region is used as the seed? The reason for these differences is not obvious, and therefore should be addressed.

One likely explanation for the distinct relationship between regional activity and behavior is that parietal and frontal cortices play distinct roles in verbal WM. A large amount of evidence supports different contributions, which would be expected to influence the effects of brain stimulation on reaction times. For example, evidence suggests that the regions have different contributions to phonological storage and executive control (Paulesuet al., Nature 1993; 362(6418): 342-5).

Similar relationships between parietal BOLD activity and reaction times have been reported before in working memory tasks. For example, Tomasi et al. observed a correlation between reaction times and BOLD signal in a similar task to ours (Neuroimage 2011; 54(4): 3101-10), while Honey et al. showed a reaction time-BOLD relationship in bilateral posterior parietal cortex (Honeyet al., Neuroimage 2000; 12(5): 495-503). Moreover, in a recent study resting state functional connectivity density in the right parietal lobule negatively correlated with RT for the 2-back task (Liuet al., Behav Brain Res 2017; 316: 66-73). Overall these findings link activity in the parietal cortex to reaction time on working memory tasks.

The possible explanations for the differential effects of tACS on frontal and parietal regions has now been extended in the Discussion section of the manuscript:

“The fact that a relationship with response times was observed for the parietal but not the frontal region might be explained by the different roles attributed to parietal and frontal cortices in verbal WM. […] Thus, by increasing neural activity in the parietal region, tACS might have interacted with the physiological mechanisms associated with response production.”

We also observed distinct effects of stimulation on parietal lobe functional connectivity. There are a number of possible explanations for this. It is possible that the ability to influence functional connectivity by stimulation might vary depending on the baseline state of a brain region. This is likely to vary across brain regions, which would therefore influence the effects of stimulation. For example, if frontal functional connectivity was already operating at peak levels during task periods without stimulation and could not be further modulated by tACS. A more intriguing possibility is that distinct frequency channels are responsible for carrying feedforward and feedback signalling. Such a distinction has been observed in the visual cortex, where feedforward influences are carried by theta and gamma band synchronization, while feedback influences by beta band synchronization (Bastoset al., Neuron 2015; 85(2): 390-401). In this framework it is possible that tACS applied at the theta frequency would differentially enhance feedforward connectivity across the stimulated fronto-parietal network. Further work will be needed to explore this question, for example by combining tACS with electrophysiological methods such as electrocorticography.

We now include a discussion of these points in the Discussion section of the manuscript:

“This pattern was observed when the parietal region was used as a seed, but not for the frontal region. […] Studies combining tACS with electrophysiological methods could help explore this hypothesis.”

6) Discussion: Would the authors care to speculate how findings from forms of non-invasive stimulation, e.g., tDCS, relate to the current data? It seems as if this other literature would be relevant to the current Discussion and that it would add breadth to the interested audience.

We thank the reviewers for this suggestion. We have added a paragraph to the Discussion to elaborate on how other forms of non-invasive brain stimulation, particularly tDCS could modulate working memory processes and how those relate to the current data:

“Other studies have shown that additional forms of non-invasive transcranial electrical stimulation (tES), particularly transcranial direct current stimulation (tDCS) modulate WM performance. […] A study comparing different tES modalities could help answer this question. Such a study would benefit from applying a similar methodology to the one we employed, in which blocks of short durations of different tES modalities could be combined with fMRI.”